# Amyloidosis in Childhood: A Review of Clinical Features and Comparison with Adult Forms

**DOI:** 10.3390/jcm13226682

**Published:** 2024-11-07

**Authors:** Giovanni Battista Zamarra, Marina Sandu, Nicholas Caione, Gabriele Di Pasquale, Alessio Di Berardino, Armando Di Ludovico, Saverio La Bella, Francesco Chiarelli, Valentina Cattivera, Jacopo Colella, Giulio Di Donato

**Affiliations:** 1Department of Pediatrics, L’Aquila University—UNIVAQ, 67100 L’Aquila, Italy; giovannibattista.zamarra@graduate.univaq.it (G.B.Z.); marina.sandu@graduate.univaq.it (M.S.); nicholas.caione@graduate.univaq.it (N.C.); gabriele.dipasquale1@graduate.univaq.it (G.D.P.); alessio.diberardino1@graduate.univaq.it (A.D.B.); valentina.cattivera@graduate.univaq.it (V.C.); jacopo.colella@graduate.univaq.it (J.C.); 2Department of Pediatrics, “G. D’Annunzio” University, 66100 Chieti, Italy; armandodl@outlook.com (A.D.L.); saverio.labella@studenti.unich.it (S.L.B.); chiarelli@unich.it (F.C.)

**Keywords:** amyloidosis, pediatrics, misfolded proteins

## Abstract

Amyloidosis is a rare multisystem disorder characterized by extracellular accumulation of insoluble fibrils in various organs and tissues. The most common subtype in the pediatric population is systemic reactive amyloidosis, typically developing secondary to chronic inflammatory conditions and resulting in deposition of serum amyloid A protein in association with apolipoprotein HDL3. Clinical presentation is highly variable and is mostly influenced by specific organs involved, precursor protein type, and extent of amyloid deposition, often closely reflecting clinical features of the underlying disease. The most critical determinants of prognosis are cardiac and renal involvement. Diagnosis of amyloidosis is confirmed by tissue biopsy, which remains the gold standard, followed by precise amyloid fibril typing. The primary therapeutic approach is directed towards controlling underlying disease and reducing serum levels of precursor proteins to prevent further amyloid deposition. This study aims to highlight the main clinical characteristics of amyloidosis with onset in childhood, emphasizing the key differences compared to adult form.

## 1. Introduction

Amyloidosis is a rare multisystemic disorder characterized by the extracellular accumulation of insoluble fibrils formed from misfolded protein subunits (from 10 to 15 kD). These fibrils (around 10 nm in diameter) are constituted of peptides arranged in an antiparallel β-sheet structure, conferring insolubility, resistance to proteolysis, and specific dyeing properties, such as Congo red affinity and apple-green birefringence under polarized light microscopy, in addition to a distinctive fibrillar appearance under electron microscopy [1,2,3].

Amyloid fibrils are made up of additional components, including serum amyloid P (SAP), a calcium-dependent glycoprotein that improves fibril stability, glycosaminoglycans (GAGs) that promote nucleation, and apolipoprotein E (APOE) [2].

Amyloidosis can be classified as localized or systemic. Localized amyloidosis is confined to a single organ or tissue, as may be seen in Alzheimer’s disease and type II diabetes, while systemic amyloidosis involves a lot of organs and tissues. Systemic amyloidosis is categorized into four main types: AL, AA, ATTR, and Aβ2M, with additional rare variants [2].

The most common type in adults is light-chain amyloidosis (AL), often associated with low-grade clonal proliferations, multiple myeloma, or, less frequently, non-Hodgkin lymphomas or Waldenström’s disease. In this kind of amyloidosis, the precursor proteins are immunoglobulin light chains (κ o λ), leading to a wide range of clinical manifestations [2].

The predominant form in children and the second most common type in adults is systemic AA amyloidosis [3]. It derives from the acute-phase protein Serum Amyloid A (SAA), associated with Apolipoprotein HDL3, and is frequently related to chronic inflammatory conditions. Hereditary periodic fever syndromes, such as familial Mediterranean fever (FMF) and other autoinflammatory disorders, are the most common causes of amyloidosis in pediatric patients. Other pediatric causes include systemic juvenile idiopathic arthritis (JIA) [4] and various chronic inflammatory conditions. In contrast, AA amyloidosis does develop very rarely as a complication of other chronic inflammatory diseases in adults and almost never in children, such as Sjogren’s syndrome [5]. Currently, the prevalence of secondary amyloidosis is significantly reduced for advances in biological therapies [2,3,6].

The list of conditions associated with AA amyloidosis is continually expanding; a significant susceptibility factor for idiopathic form is obesity, as suggested by recent evidence [7].

The third most common type is transthyretin amyloidosis (ATTR). This form is primarily hereditary, resulting from point mutations in transthyretin (TTR) protein, leading to peripheral and autonomic neuropathy. Senile systemic amyloidosis represents an acquired form and usually presents in elderly men with slowly progressive cardiomyopathy. Finally, most frequently associated with long-term dialysis in end-stage renal disease is β2-microglobulin amyloidosis (Aβ2M), though a hereditary form has also been identified, characterized by autonomic neuropathy and gastrointestinal involvement [2].

## 2. Epidemiology

A recent Swedish study estimated the incidence of AL amyloidosis at around 3.2 cases per million people per year, while the reported incidence of AA amyloidosis was 2.0 cases per million annually [2]. AL amyloidosis is more frequently observed in men, while AA amyloidosis is more common in women, largely due to the higher prevalence of rheumatoid arthritis in the female population. The median age of presentation for both types of amyloidosis is typically between 55 and 60 years. Both forms of amyloidosis are characterized by a poor prognosis, with median survival ranging from 6 to 12 months for AL amyloidosis and 3 to 4 years for the AA variant [2].

Amyloidosis is extremely rare in children [8], with its incidence largely reflecting the frequency of predisposing conditions. FMF primarily affects populations in the Mediterranean region, including Turkey, Arabia, Armenia, and non-Ashkenazi Jewish communities [9]. Before the large use of colchicine in therapy, amyloidosis was a frequent complication of FMF, affecting 60–75% of patients over the age of 40. Many studies have demonstrated variations in FMF severity based on geographic region, with a significantly higher incidence of amyloidosis observed among Turkish and Armenian individuals residing in their countries of origin compared to those who have emigrated to Northern Europe or the United States, suggesting the importance of environmental factors in the exacerbation of the inflammatory response [9].

The incidence of reactive amyloidosis for Tumor Necrosis Factor Receptor-Associated Periodic Syndrome (TRAPS) is reported to range from 14% to 25%, while it is approximately 3% in Mevalonate kinase deficiency (MKD) [3]. The most common rheumatic disease in childhood is JIA, and the historical incidence of AA amyloidosis in JIA patients, which varied from 1% to 10%, has substantially reduced due to earlier diagnosis and improved treatment strategies [3].

## 3. Classification and Immunopathophysiology

In accordance with guidelines established by the International Society of Amyloidosis (ISA), amyloid is identified as an extracellular aggregation of fibrillary proteins, that shows a Congo red affinity and apple-green birefringence under polarized light microscopy [10]. Amyloid formation arises from several factors: increased concentration of proteins with acquired or hereditary mutations [11], wild-type proteins with an intrinsic propensity for misfolding [12], or proteolytic remodeling of wild-type proteins. 

Normally, a robust quality control system guarantees correct protein folding and eliminates misfolded proteins at both intracellular and extracellular levels. Amyloidogenesis occurs when this system is overwhelmed or compromised, particularly with aging [13,14,15]. During amyloid fibril formation, the amyloid P component, apolipoprotein E (APOE), and glycosaminoglycans (GAGs) contribute to the formation and persistence of amyloid deposits. These components are consistent across all amyloid types, regardless of the specific protein involved, making them true markers of amyloidosis. The current classification of amyloidosis is based on type of amyloid protein, using the standard nomenclature “AX”, where “A” stands for amyloidosis and “X” represents the protein in the fibrils [10] (Figure 1).

To date, 36 amyloidogenic proteins have been identified in humans. Amyloidosis can be classified into localized and systemic forms, depending on the degree of fiber deposition. Localized amyloidosis is associated with at least 19 proteins, while systemic amyloidosis involves 14 proteins, with some variants [10].

Amyloidosis may be classified as hereditary or acquired. The ISA recommends the term “hereditary” over “familial” for amyloidosis involving mutated proteins. “ATTRv” (variant) is preferred over “ATTRm” (mutant) to describe hereditary transthyretin amyloidosis, while “ATTRwt” refers to wild-type transthyretin amyloidosis [10]. Although some phenotypes are associated with specific amyloid types, classification based on clinical presentation alone is not recommended due to heterogeneity influenced by genetic and environmental factors [10]. The “amyloid hypothesis” postulates that amyloid fibril formation involves multiple steps: unfolding and misfolding of precursor proteins, nucleation, polymerization, fibril elongation, and tissue deposition. Normally, protein folding is mediated by chaperones in the endoplasmic reticulum, but extracellular factors such as low pH, oxidative stress, and elevated temperature can induce protein unfolding. For amyloidogenesis, precursor proteins must adopt a partially unfolded conformation. Misfolded proteins that escape degradation can aggregate into amyloid fibrils.

Amyloid fibril formation occurs in two phases: nucleation and elongation. During the nucleation phase, misfolded protein monomers form insoluble oligomers, which are the core of fiber formation. This step is energy-intensive and rate-limiting. Intermediate oligomers are highly toxic, generating reactive oxygen species and inducing cellular stress Once the nucleus is formed, fibrils grow by nucleation-dependent polymerization, adding more monomers [16,17,18,19,20,21,22].

Regardless of the precursor protein, amyloid fibrils share structural characteristics: nonbranching, insoluble fibrils with a diameter of 7.5–10 nm, low molecular weight (5–25 kDa), and a cross-β-sheet secondary structure [22]. Intermolecular hydrogen bonds between the amide and carbonyl groups of the main chain monomers stabilize the β-sheet structure. Protofibrils stack perpendicularly to the fiber axis to form amyloid fibrils [23]. Organ tropism in amyloidosis is influenced by the type and sequence of the precursor protein, but the exact mechanism is still unknown. Fibril formation requires specific tissue conditions, including adequate local amyloid concentration and a conducive environment, such as low pH. Factors like the extracellular matrix, proteases, shear forces, and metals also promote aggregation and fibril formation [24].

### 3.1. Forms of Amyloidosis

Table 1 encompasses the main forms of amyloidosis currently recognized. The following discussion will focus on the most frequently documented phenotypes in the literature.

#### 3.1.1. AL Amyloidosis

AL amyloidosis is the most prevalent form of systemic amyloidosis in developed countries [25,26,27], primarily caused by clonal proliferation of bone marrow plasma cells that secrete a monoclonal immunoglobulin light chain, which deposits as amyloid fibrils in tissues. Misfolding may result from primary sequence of monoclonal light chain or from genetic and epigenetic factors. AL amyloidosis is often associated with multiple myeloma and other B-cell lymphoproliferative disorders, such as non-Hodgkin lymphoma and Waldenström macroglobulinemia. Hereditary AL amyloidosis is extremely rare, particularly in children [11,28,29].

#### 3.1.2. AA Amyloidosis

AA amyloidosis arises from serum amyloid A (SAA) protein, an acute-phase reactant produced by the liver [30], typically associated with chronic inflammatory conditions. It is the leading cause of systemic amyloidosis in developing countries due to high prevalence of infections like subacute bacterial endocarditis and tuberculosis [31,32,33,34,35]. Other causes are JIA, IBD, tumor, and vasculitis. AA amyloidosis is also the most common form in pediatric populations, frequently associated with hereditary autoinflammatory diseases. Hereditary AA amyloidosis is linked to autoinflammatory diseases, while hereditary amyloidoses involve mutations in amyloidogenic protein itself. Specific risk factors have been identified in AA amyloidosis development in autoinflammatory disease, and these are explained in Table 2 [36].

However, high levels of SAA resulting from continuous subclinical or clinical inflammation are the leading cause of AA amyloidosis development [36]. Indeed, chronic inflammation may be observed in up to one-third of children with FMF during intercritical times, causing an elevated risk of renal AA amyloidosis over time. Recently, obesity has emerged as a significant risk factor for idiopathic AA amyloidosis [37]. Notably, 19% of AA amyloidosis cases lack an identifiable underlying condition [2,3,6,7]. Although chronic inflammation is a key factor in the development of AA amyloidosis, it does not affect all patients with chronic diseases. Serum amyloid A, the precursor protein, is produced by the liver in response to proinflammatory cytokines such as IL-6, IL-1, and TNF-α, and plays a role in lipid metabolism. Under normal conditions, SAA is fully degraded by macrophages through lysosomes. In amyloidosis, however, incomplete degradation and the accumulation of intermediate products occur, leading to amyloid deposition in tissues [38,39]. Moreover, there is growing evidence that genetic predisposition may lead to higher susceptibility to accumulation and deposition of SAA, leading to extracellular accumulation and damage.

#### 3.1.3. ALECT2 Amyloidosis

ALECT2 amyloidosis, the third most common renal amyloidosis after AL and AA [40], involves amyloid fibrils derived from leukocyte chemotactic factor 2 (LECT2) [41] Although no genetic component has been identified, there is a strong association with Mexican–American ethnicity, as well as among Native Americans, Punjabis, Egyptians, and First Nations peoples of British Columbia [33,40,42,43,44]. ALECT2 primarily affects older patients.

#### 3.1.4. Hereditary Amyloidosis

Hereditary amyloidoses are autosomal dominant diseases with variable penetrance and onset, often lacking a family history. These conditions involve variant plasma proteins forming amyloid deposits, typically beginning in mid-life. Collectively, they account for about 10% of all systemic amyloidosis cases [45,46,47]. In isolated areas of Portugal, Sweden, and Japan, a founder effect has led to higher incidences [48]. ATTRv is the most common hereditary amyloidosis globally, caused by mutations in the TTR protein. TTR, primarily produced by the liver, transports thyroxine and retinol. Mutations in the TTR gene weaken monomeric interactions, leading to tetramer dissociation and amyloid fibril accumulation. The Val122Ile mutation, prevalent in 3–4% of African Americans, leads to a cardiac phenotype, whereas Val30Met [27,49], common in Sweden, Portugal, and Japan, is associated with polyneuropathy [50,51].

#### 3.1.5. Iatrogenic Amyloidosis

Aβ2M amyloidosis results from the accumulation of β2-microglobulin, associated with rheumatological manifestations in long-term hemodialysis patients. β2-microglobulin is normally excreted by kidneys, and its levels increase in end-stage renal disease, as it is above the filtration threshold of some dialysis membranes [11,52].

## 4. Clinical Manifestations

Amyloidosis is a multisystemic disorder with various clinical manifestations that depend on the precursor protein, the extent of amyloid deposition, and organs involved (Table 3 and Table 4) [46].

Like the adult population, in pediatric patients, symptoms due to amyloid deposition may also present with clinical manifestations of an underlying disease, which is often different to that in adults. Commonly affected organs include heart, kidneys, nervous system, liver, and gastrointestinal tract (Figure 2). Nonspecific symptoms such as fatigue and weight loss may develop gradually and often go unnoticed until appearance of more specific and alarming symptoms [1,2].

In a study of 459 adult patients with amyloidosis, only 26% received a diagnosis within one year of symptom onset, and 49% saw four or more physicians before a definitive diagnosis was made [53]. Although analogous data for pediatric population are not available, it is reasonable to assume that diagnostic delays may largely overlap. Specific signs of AL amyloidosis, such as tongue enlargement and periorbital purpura, occur in only 15% of patients [26]. Cardiac and renal involvement are the most important predictors influencing survival [54]. Amyloidosis should be considered in the differential diagnosis of adult nondiabetic nephrotic syndrome; heart failure with preserved ejection fraction; unexplained hepatomegaly without imaging abnormalities; peripheral neuropathy; and monoclonal gammopathy of undetermined significance (MGUS) with atypical clinical features [55]. In AL amyloidosis, the heart is involved in about 75% of cases, the kidneys in 57%, the nerves in 22%, and the liver in 20% [56].

### 4.1. Kidney

Renal parenchyma is one of the most frequently involved sites in systemic and hereditary amyloidosis [40]. In adults, renal amyloidosis typically presents with nonselective proteinuria and/or renal insufficiency [46], commonly involving glomeruli and requiring renal biopsy [7]. Amyloid is suspected through the presence of amorphous “hyaline” deposits, which are weakly periodic acid–Schiff (PAS)-positive and exhibit argyrophilic loss [40,57,58]. There are, on the other hand, cases in which patients have only mild proteinuria and normal renal function [59]. In children, renal amyloidosis can present with proteinuria or nephrotic syndrome, which may be followed by chronic renal insufficiency. As in the adult population, even in children the type of proteinuria is not specified. Asymptomatic proteinuria is the most common initial presentation of kidney disease; therefore, urine examination should be performed in all patients with disease associated with a risk of secondary amyloidosis [1]. Unlike in adults, hematuria and hypertension rarely occur in children [54]. Since primary amyloidosis is extremely rare in children, secondary amyloidosis is the most common pattern of renal amyloidosis in pediatric patients [6]; in adults, however, the renal pattern may result from either primary or secondary amyloidosis.

### 4.2. Heart

Cardiac amyloidosis is a rare cause of cardiomyopathy, reported exclusively in adults. Main systemic amyloidosis with clinically significant cardiac involvement includes AL and ATTR [50], while AA amyloidosis rarely affects heart clinically [7,60,61]. Cardiac presentation in AL amyloidosis in adults usually includes heart failure with preserved ejection fraction [62,63]. Early symptoms may include dyspnea, edema, hypotension, arrhythmias, and atypical angina due to amyloid deposits in coronary arteries [2]. Ventricular wall thickening with preserved ejection fraction and absence of left ventricular dilation may suggest amyloidosis [64]. In childhood, we find only a case report about a 12-year-old boy presented with a single syncopal event, intermittent and nonspecific chest pain, and fatigue and shortness of breath, which was diagnosed with ventricular noncompaction by echocardiography, and he has preserved ejection fraction. Eventual genetic testing confirmed a TTR gene mutation associated with hereditary transthyretin amyloidosis [65]. This case report could provide a basis for understanding the early symptomatology of cardiac amyloidosis in pediatrics, which may present with early, nonspecific symptoms such as syncope, intermittent and nonspecific chest pain, fatigue, and shortness of breath. However, further studies are needed for a complete understanding.

### 4.3. Gastrointestinal System

The gastrointestinal tract is frequently involved in systemic amyloidosis, most commonly AL, followed by AA and hereditary types [7]. Gastrointestinal involvement occurs in about 17% of AL amyloidosis patients [55] and 20% of those with reactive amyloidosis [66]. Clinical presentations in adults can range from vague symptoms such as epigastric pain, diarrhea, malabsorption, hematochezia, hematemesis, and constipation to obstruction, gastrointestinal bleeding, and perforation [67]. In children, we found that early symptoms include epigastric pain, abdominal pain, and gastrointestinal bleeding, together with symptoms of the underlying condition, as described by Kamei et al. in a case report of a 16-year-old Japanese girl who died from secondary AA amyloidosis associated with juvenile rheumatoid arthritis; the patient presented with amyloid deposits in various sites, including the kidneys, vascular walls, spleen, and gastrointestinal tract, particularly in the ileum. This led to peritonitis and hemorrhagic necrosis of the terminal ileum, ultimately resulting in her death [8]. From this, we can deduce that symptoms of gastrointestinal amyloidosis are similar in adults and children. Additionally, in children, symptoms of an underlying condition will also be present. Deposition of amyloid in the liver and spleen in adult causes hepatomegaly and splenomegaly, with elevation of liver enzymes, which can complicate drug management; furthermore, splenomegaly can lead to splenic rupture [68,69]. In children, however, the presence of hepatomegaly is not necessarily associated with elevated liver enzymes, which may sometimes remain within normal ranges [70].

### 4.4. Lungs

In lungs, amyloid deposits can be localized or associated with systemic disease [71,72]. Lung biopsy is rarely performed. Localized disease like nodular amyloidosis carries a better prognosis compared to diffuse alveolar amyloidosis [73]. Pulmonary amyloidosis may present radiologically with multiple pulmonary nodules, a solitary nodular mass, or ground-glass opacities, and symptoms such as dyspnea, physical decline, cough, hemoptysis, and bronchitis [71]. There are no reported cases in the literature regarding amyloid deposition in the lungs of children.

### 4.5. Nervous System

Sensory and motor peripheral neuropathy and autonomic neuropathy can occur in amyloidosis, although they are rare in secondary forms [1]. Amyloidosis is a relatively rare cause of peripheral neuropathy, accounting for approximately 3% of cases in the Mayo Clinic laboratory [74,75]. Diagnosis is often delayed because clinical features can mimic many other neuropathies [7]. No cases were found in children, but it could plausibly occur in children as well.

## 5. Diagnosis

Amyloidosis should be considered in the differential diagnosis of patients with nephrotic proteinuria, heart failure with preserved ejection fraction, neuropathy, hepatomegaly, and unexplained gastrointestinal symptoms [25,26,76]. The diagnostic approach begins with routine tests to assess organ involvement, including renal and liver function tests, chest X-rays, spirometry, ECG, BNP, and troponin. A definitive diagnosis is confirmed through biopsy of affected tissues and amyloid typing. The gold standard for diagnosis involves detecting amyloid deposits in tissue samples [57,77,78]. Under optical microscopy, these deposits appear extracellular, eosinophilic, and metachromatic. Congo red staining, viewed under polarized light, is a reference standard, revealing characteristic apple-green birefringence due to alternating hydrophobic and hydrophilic regions along the fiber axis [57,79]. Identifying a specific amyloid protein type is crucial, as it impacts treatment decisions [11,57]. While biopsies of target organs such as the kidney or heart are the most sensitive, less invasive options like abdominal fat or bone marrow biopsies are often recommended [57,79,80,81,82,83]. Sensitivity of fat biopsies for amyloid detection varies by amyloid type, ranging from 70 to 90% for AL amyloidosis and 67% for hereditary ATTR amyloidosis to only 14% for wild-type ATTR amyloidosis [57,80,83,84]. Sensitivity for bone marrow biopsies is approximately 60% [25,79,85].

Recently, pyrophosphate scintigraphy has been proposed for the radiological detection of ATTR cardiomyopathy [77]. Confirming the presence of amyloid is just the beginning; determining a specific type of amyloid is essential for appropriate treatment planning. 

Immunofluorescence and immunohistochemistry characterization use antibodies against various amyloidogenic proteins. Immunohistochemistry employs antibodies conjugated with enzymes (such as horseradish peroxidase) to visualize the presence of target antigens in tissues. Enzymatic reaction products result in colorimetric changes, facilitating observation of structures under an optical microscope [86]. Immunofluorescence identifies the same antigens through antibodies conjugated to fluorochromes that emit light when exposed to ultraviolet light, enabling their observation under a fluorescence microscope [87]. A panel of antibodies directed against precursor fibrils should be ensured to facilitate the identification of the main forms of amyloidosis. For instance, antibodies such as anti-AA, anti-ALλ, anti-ALκ, anti-ATTR, anti-AHγ, anti-Aβ2M, anti-AFib, anti-ALECT2, anti-AApoAI, anti-ALys, anti-ACys, and anti-AGel should be included. Currently, the literature remains uncertain about which of the two techniques offers superior sensitivity and specificity for diagnostic purposes [40,88,89,90,91,92]. In challenging cases where immunofluorescence or immunohistochemistry are inconclusive, or if less common types are detected [40], proteomic techniques allow for unequivocal identification of protein-forming amyloid deposits, particularly utilizing mass spectrometry [25,26,27,50,88,89,93,94,95,96,97,98,99,100]. Molecular and genetic testing can be performed in cases of hereditary amyloidosis (e.g., TTR, fibrinogen, lysozyme, apolipoproteins AI and AII, and gelsolin) [27,76,101,102,103,104].

## 6. Treatment and Prognosis

Amyloidosis has a poor prognosis if untreated and undiagnosed. Therefore, early diagnosis and starting treatment as early as possible are necessary to stabilize disease and prevent ongoing progression. The primary strategy is to reduce inflammatory activity of underlying condition, and it is important to stop the growth of amyloid deposits by eliminating the precursors necessary for their formation. Although there is a lack of longitudinal studies regarding pediatric population, it can be assumed that, compared to the adult/elderly population, which is generally burdened by greater comorbidities, treating the underlying condition may ensure a better prognosis and a lower risk of disease development. Treatment of amyloidosis includes treatment of the underlying cause, symptom management, chemotherapy, peripheral blood stem cell transplant, organ transplant, physiotherapy, and psychosocial support.

### 6.1. AA Amyloidosis

For AA amyloidosis, treatment aims to reduce inflammation, thus maintaining serum amyloid A levels to baseline values (below 3 mg/L) [105]. Keeping SAA levels below 10 mg/L can increase the 10-year survival rate to 90%, while levels above 10 mg/L are associated with a survival rate of less than 40% [106]. These are data concerning adults; for the pediatric population, there are no data. To achieve normal SAA values there must be complete suppression or eradication of the underlying chronic inflammatory disease. Improvement in treatments for chronic inflammatory diseases with the introduction of anti-inflammatory drugs like methotrexate and biologics targeting TNF and IL-1 have made the suppression of SAA at low or even normal serum concentrations a realistic goal [107]. In addition to colchicine, new monoclonal antibodies such as IL-1 inhibitors anakinra and canakinumab, and IL-6 inhibitor tocilizumab, may optimize the treatment of FMF and the prevention or control of AA amyloidosis [108]. It should be noted that even heterozygous FMF genotypes have an increased risk of AA amyloidosis, and treatment should also be administered in paucisymptomatic patients to prevent complications [109]. Moreover, no agreement has been established on the discontinuation time for colchicine in asymptomatic FMF patients, with most patients subjected to lifelong prophylaxis [109]. The previously mentioned drugs, such as methotrexate, colchicine, anakinra, canakinumab, and tocilizumab, are used in pediatrics as therapies for various conditions, including mevalonate kinase deficiency, juvenile idiopathic arthritis, cryopyrin-associated periodic syndromes, tumor necrosis factor receptor-associated periodic syndrome, and familial Mediterranean fever, which we previously noted increases the risk of pediatric amyloidosis.

### 6.2. AL Amyloidosis

AL amyloidosis is most frequently encountered in the elderly population, with a mean age of diagnosis of approximately 60 years and an incidence that increases with age [110]. In AL amyloidosis, the goal is to treat underlying plasma cell dyscrasia with chemotherapy and normalize kappa or lambda light chain levels [111]. High-dose melphalan (HDM) followed by autologous stem cell transplantation (ASCT) in eligible patients has shown significant benefits [112]. However, recent studies suggest that this approach may not be superior to standard-dose melphalan plus dexamethasone [113]. Meanwhile, numerous studies on new drugs such as thalidomide, bortezomib, lenalidomide, and pomalidomide have shown promising effects, often with the best results in combination with dexamethasone [114]. It has been found that in the first 6 months after diagnosis, 25% of AL amyloidosis patients die due to end-stage organ failure [56]. Several risk models are used for AL amyloidosis [115]. The Mayo Clinic 2004 model classifies patients into three stages based on cardiac troponin T below 0.035 μg/L and N-terminal probrain natriuretic peptide (NT-proBNP) below 332 ng/L. Patients are considered stage I if both markers are below thresholds, stage II if only one marker is below threshold, and stage III if both are at or above thresholds. The 2012 Mayo model classifies patients based on cardiac troponin T or high-sensitivity troponin T, NT-proBNP, and difference between involved and uninvolved free light chains. All models have similar predictive value for survival [116,117,118]. The 2004 model better predicts early mortality, while the 2012 model has a better prediction for long-term survival. AL amyloidosis prognosis can be determined by two or three blood tests [119]. AL amyloidosis is extremely rare in children [11,28,29] and we did not find data in the literature about treatment except for a case report concerning a 13-year-old girl with primary localized nasopharyngeal AL amyloidosis where the main treatment was surgical [120].

### 6.3. TTR Amyloidosis

Hereditary ATTR amyloidosis treatment has traditionally involved liver transplantation to remove source of circulating mutant TTR [121]. However, this approach may not be effective in some cases, particularly with late-onset disease or non-TTR-Met30 mutations. Wild-type ATTR amyloidosis has a median overall survival of 3.6 years, with survival rates varying by stage [121,122,123]. Treatment now available includes TTR stabilizers, such as Tafamidis, and TTR amyloid removal, as well as gene silencing, while gene editing therapies are on the way. Since TTR amyloidosis is hereditary, early diagnosis is important for improving prognosis. In endemic countries where ATTRv is often familial, family members are offered genetic testing, and presymptomatic carriers are regularly seen in clinic. Symptomatic carriers often undergo biopsy to confirm tissue deposition of TTR. In nonendemic countries, both genetic testing and testing for amyloidosis are indicated in suspected cases [124]. The only case of TTR amyloidosis we found in the literature in the pediatric field is that of cardiac TTR amyloidosis, which was primarily treated with implantation of a defibrillator and will be followed up every 3–5 years to determine when to start therapy with Tafamidis [65]. Currently, there are no studies on Tafamidis in pediatrics.

### 6.4. Ab2M Amyloidosis

In Ab2M amyloidosis, kidney transplantation is the treatment of choice. After transplantation, amyloidosis appears to stabilize, with serum β2-microglobulin levels dropping to normal, but there have been no reported decreases in amyloid deposits [125]. No data were found regarding children.

### 6.5. Other Variants

For other variants of amyloidosis, the current basis for treatment is the so-called precursor–product concept. The central idea of this concept, as described above, is that further growth of amyloid deposits will stop when the supply of the necessary precursors is interrupted [2].

## 7. Discussion

In adults, amyloidosis is most associated with conditions like multiple myeloma or chronic inflammatory diseases, where overproduction of certain proteins leads to their misfolding and deposition as amyloid. The most common form in adults is AL amyloidosis, resulting from clonal proliferation of plasma cells. This form typically affects older individuals and is associated with organ involvement, such as kidneys, heart, liver, and the nervous system. Symptoms can include fatigue, edema, weight loss, and organ-specific dysfunctions like nephrotic syndrome, cardiomyopathy, or neuropathy.

In contrast, pediatric amyloidosis is much rarer and often has a genetic basis. The most common forms in children are AA amyloidoses, such as those caused by mutations in genes associated with autoinflammatory syndromes, like FMF. These genetic mutations lead to abnormal production of amyloidogenic proteins that deposit in various tissues. Clinical presentation in children may include recurrent fevers, abdominal pain, arthritis, and, in the case of FMF, a high risk of developing renal amyloidosis. Pediatric patients may also present with systemic involvement, but progression can vary significantly depending on underlying genetic disorder.

In adults, diagnosis often involves a combination of clinical evaluation, laboratory tests, imaging studies, and tissue biopsy to confirm amyloid deposition and to determine amyloid type. In children, diagnosis is often guided by the identification of underlying genetic mutations through genetic testing, especially in cases of hereditary amyloidosis. Tissue biopsy is less commonly performed in children unless absolutely necessary.

In adults with AL amyloidosis, the primary treatment focuses on targeting underlying plasma cell disorder with chemotherapy, including drugs like bortezomib, lenalidomide, and dexamethasone, which aim to reduce the production of amyloidogenic light chains. In cases of familial amyloidosis in adults, liver transplantation may be considered to prevent further production of the abnormal protein.

In pediatric amyloidosis, treatment often focuses on managing underlying genetic conditions. For example, in FMF, colchicine is a standard treatment to prevent attacks and reduce the risk of amyloid deposition. In hereditary forms linked to other genetic mutations, emerging therapies targeting specific molecular pathways or gene therapy may offer future potential, though these treatments are still under investigation.

In conclusion, while amyloidosis shares common pathological features across age groups, differences in etiology, clinical presentation, and treatment between adults and children necessitate tailored approaches to management. Early diagnosis is crucial for initiating specific treatment for the type of amyloidosis, especially in the pediatric population, which has a longer life expectancy and fewer comorbidities, allowing for better tolerability of the various therapeutic approaches. Understanding this issue is crucial for improving outcomes and quality of life for patients.

## Figures and Tables

**Figure 1 jcm-13-06682-f001:**
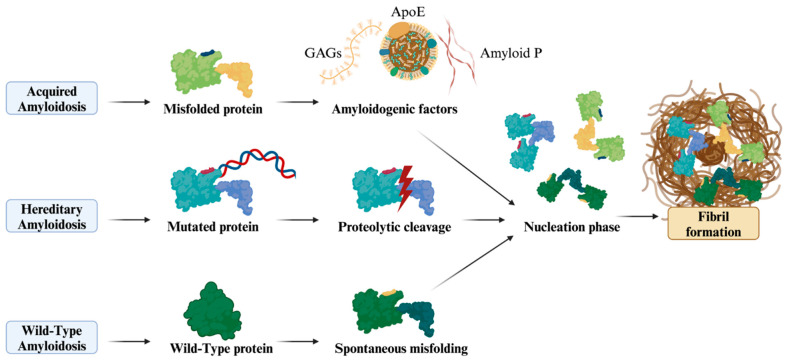
Pathways of amyloidosis: An integrated view of amyloid formation. This figure illustrates the three primary pathways of amyloid fibril formation in amyloidosis: acquired, hereditary, and wild type. In acquired amyloidosis, misfolded proteins aggregate, influenced by amyloidogenic factors such as amyloid P, ApoE, and GAGs. In hereditary amyloidosis, mutated proteins undergo proteolytic cleavage, leading to amyloid formation. In wild-type amyloidosis, native proteins spontaneously misfold, particularly with aging. Despite different origins, all pathways converge in the nucleation phase, resulting in the formation of insoluble amyloid fibrils, characterized by a cross-β sheet structure.

**Figure 2 jcm-13-06682-f002:**
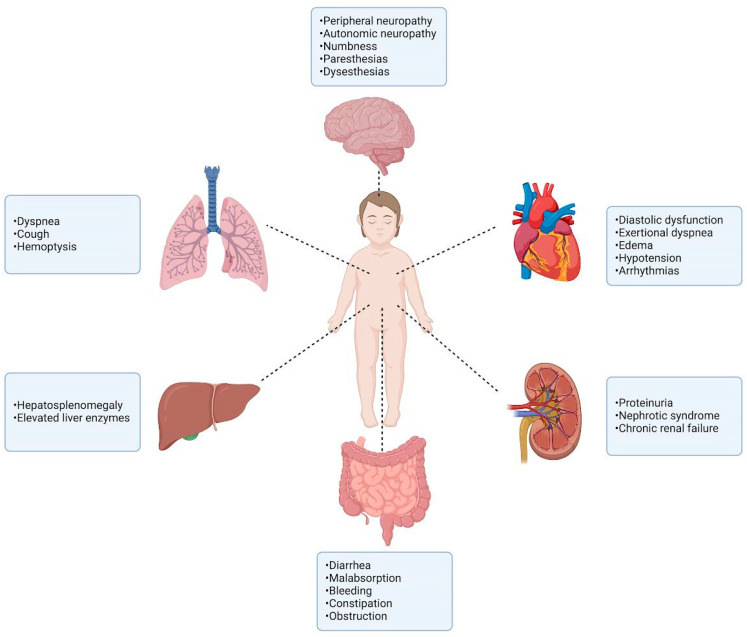
Clinical manifestations of amyloidosis.

**Table 1 jcm-13-06682-t001:** Amyloid fibril proteins and their precursors [10].

Fibril Protein	Precursor Protein	Systemic and/or Localized	Acquired or Hereditary
AL	Immunoglobulin light chain	S, L	A, H
AH	Immunoglobulin heavy chain	S, L	A
AA	(Apo) Serum amyloid A	S	A, H
ATTR	Transthyretin, wild type Transthyretin, variants	S S	A H
Aβ2M	β2-microglobulin, wild type β2-microglobulin, variants	S S	A H
AApoAI	Apolipoprotein A I, variants	S	H
AApoAII	Apolipoprotein A II, variants	S	H
AApoAIV	Apolipoprotein A IV, wild type	S	A
AApoCII	Apolipoprotein C II, variants	S	H
AApoCIII	Apolipoprotein C III, variants	S	H
AGel	Gelsolin, variants	S	H
ALys	Lysozyme, variants	S	H
ALECT2	Leukocyte chemotactic factor-2	S	A
AFib	Fibrinogen α, variants	S	H
ACys	Cystatin C, variants	S	H
ABri	ABriPP, variants	S	H
ADan	ADanPP, variants	L	H
Aβ	Aβ protein precursor, wild type Aβ protein precursor, variant	L L	A H
AαSyn	α-Synuclein	L	A
ATau	Tau	L	A
APrP	Prion protein, wild type Prion protein variants Prion protein variant	L L S	A H H
ATMEM106B	Transmembrane 106B (TMEM106B)	L	A
ACal	(Pro)calcitonin	L	A
AIAPP	Islet amyloid polypeptidec	L	A
AANP	Atrial natriuretic peptide	L	A
APro	Prolactin	L	A
ASom	(Pro)somatostatin	L	A
AGluc	Glucagon	L	A
APTH	Parathyroid hormone	L	A
AIns	Insulin	L	A
AEnf	Enfurvitide	L	A
AGLP1	Glucagon-like peptide 1 analog	L	H
AIL1RAP	Interleukin-1 receptor antagonist protein	L	A
ASPC	Lung surfactant protein	L	A
ACor	Corneodesmosin	L	A
AMed	Lactadherin (MFG-E8)	L	A
AKer	Kerato-epithelin	L	A
ALac	Lactoferrin	L	A
AOAAP	Odontogenic ameloblast-associated protein	L	A
ASem1	Semenogelin 1	L	A
ACatK	Cathepsin K	L	A
AEFEMP1	EGF-containing fibulin-like extracellular matrix protein 1 (EFEMP1)	L	A

**Table 2 jcm-13-06682-t002:** Specific risk factors in AA amyloidosis development in autoinflammatory disease.

Disease	Risk Factors
Familial Mediterranean fever (FMF)	Male gender Early onset disease M694V missense variant in *MEFV*
Mevalonate kinase deficiency (MKD)	V377I and the I268T variants in the *MVK* gene
Cryopyrin-associated periodic syndrome (CAPS)	R260W variant in the *NLRP3* gene
Tumor Necrosis Factor Receptor-Associated Periodic Syndrome (TRAPS)	T50M variant in the *TNFRSF1A* gene

**Table 3 jcm-13-06682-t003:** Associated disease and target organs in adult and childhood amyloidosis.

Fibril Protein	Precursor Protein	Associated Disease in Adult	Associated Disease in Childhood	Target Organs
AL	Immunoglobulin light chain	Multiple myeloma Non-Hodgkin lymphoma Waldenström macroglobulinemia	Anecdotes	All organs, usually except CNS.
AA	(Apo) Serum amyloid A	Chronic infections IBD Rheumatoid arthritis Vasculitis Tumor	Autoinflammatory disease JIA Chronic infections IBD	All organs except CNS.
ATTR	Transthyretin, wild type Transthyretin, variants	Aging N/A	N/A N/A	Heart mainly in males, lung, ligaments, tenosynovium PNS, ANS, heart, eye, kidneys, leptomeninges.
ALECT2	Leukocyte chemotactic factor-2	Aging	N/A	Kidney, primarily.
Aβ2M	β2-microglobulin, wild type β2-microglobulin, variants	Hemodialysis N/A	Hemodialysis N/A	Musculoskeletal system ANS, tongue, heart.

**Table 4 jcm-13-06682-t004:** Clinical manifestations of amyloidosis.

Organ Involved	Symptoms
Kidney	Proteinuria; nephrotic syndrome; renal insufficiency; hematuria; arterial hypertension; edema.
Hearth	Heart failure with preserved ejection fraction; diastolic dysfunction; exertional dyspnea; edema; hypotension; syncope; arrhythmias; angina; conduction disturbances.
Gastrointestinal	Diarrhea; malabsorption; hematochezia; hematemesis; constipation; obstruction or pseudo-obstruction; gastrointestinal bleeding; intestinal perforation; nausea; vomiting; epigastric pain; dysphagia.
Liver and spleen	Hepatomegaly; elevated liver enzymes; splenomegaly; hyposplenism; splenic rupture.
Lung	Dyspnea; cough; hemoptysis; bronchitis.
Nervous system	Peripheral neuropathy; autonomic dysfunction; spinal stenosis/pseudoclaudication.
Others	Carpal tunnel syndrome; macroglossia, periorbital purpura; bicipital tear; hemorrhagic diathesis; atypical myeloma; bone cysts; spondyloarthropathies; pathological fractures; adrenal insufficiency; hypothyroidism; hypogonadism.

## Data Availability

Not applicable.

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
