# Peer review of "Amyloidosis in Childhood: A Review of Clinical Features and Comparison with Adult Forms"

_jcm, 2024, doi:10.3390/jcm13226682_

Round 1
Reviewer 1 Report
Comments and Suggestions for Authors
The manuscript by Zamarra, G.B. et al. reviewing amyloidosis in childhood seems. The style is fine but I think a bit more extension would be good (see minor comments). Honestly, I can find few differences between childhood amyloidosis and the adult form, but the topic is interesting.
There are some minor comments:
- Introduction, page 1, lines 35-37. This sentence is not necessarily wrong, but I think it can be improved. It´s true that Alzheimer and diabetes have been proposed as examples of localized amyloidosis, but probably other mechanisms coexist. Why don´t you include “may” before seen in line 36?
- Introduction, page 2, line 48. Are the most common causes of amyloidosis.
- I miss a subheading number 3. There are 3.1, 3.2, 3.2.1… but no subheading 3.
- Classification and immunopathophysiology, page 2, line 93. The birefringence is also observed in Congo red stained slides.
- Classification and immunopathophysiology, page 3, lines 120-126. Part of this information is already mentioned in the introduction. I think the references in this text to the ISA recommendations are a bit confusing. I think a table or diagram with the different subtypes of amyloidosis (named according to the ISA guidelines) can be good here.
- AL Amyloidosis, page 4, line 151. “is the most prevalent form”. The same in line 164.
- AA Amyloidosis, page 4, line 165. I though the term “familial amyloidosis” was not recommended by the ISA… Please, revise this sentence.
- AA Amyloidosis, page 4, line 170. In AA amyloidosis caused by mevalonate kinase deficiency, the prevalent risk factors are (...). The same for other inflammatory conditions in this paragraph. You should also consider dividing this long paragraph in several ones, to separate the different conditions.
- Forms of amyloidosis, page 4, line 149. I miss a paragraph here explaining why you only mention 5 types of amyloidosis. There are 36 types, why don´t you mention the other 31? Children aren’t affected?
- Clinical manifestations, page 6, line 226. “of an underlying disease”.
- Heart, page 8, line 264. “venthe tricular”?
- Diagnosis, page 9, line 310. I think you should mention what immunohistochemical typing is currently advised. You should recommend a panel of antibodies for the characterization. You should also explain why immunofluorescence is preferred over conventional immunohistochemistry.
- Treatment, page 9, line 326. Eliminating the precursors necessary for their formation.
- Treatment, page 9, line 328. A dot is required before “in fact”.
- Treatment, pages 9 and 10, lines 331-371. This paragraph should be divided, probably in 3 smaller paragraphs. One for AA amyloidosis, other for AL and a third one for TTR. You should also include a paragraph for the other less common variants, at least the ones described in section 3.2. Please remember that there are other forms of amyloidosis, which were not mentioned in the text.
- Conclusion, page 10, lines 373-376. This text has no place in conclusions. This is already mentioned particularly in the introduction.
- Conclusion, page 10. Actually, it seems you named “conclusions” to an actual “discussion”. Please change.
The language is generally fine. I detected various small issues, some of them mentioned as minor faults. A professional review is advised.
Author Response
First and foremost, I would like to express my gratitude for your thorough review of our work.
Below are the corrections made as you indicated.
-Introduction, page 1, lines 35-37: we include “may” before seen
-Introduction, page 2, line 48. Are the most common causes of amyloidosis
-Now "Classification and Immunopathophysiology" subheading number 3. Forms of Amyloidosis subheading 3.1
-Classification and immunopathophysiology, page 2, line 93. The birefringence is also observed in Congo red stained slides.
- Classification and immunopathophysiology, page 3, lines 120-126. We have created the current Table 1, which includes all the forms of amyloidosis listed by the ISA.
-AL Amyloidosis, page 4, line 151. “is the most prevalent form”. The same in line 164
-AA Amyloidosis, page 4, line 165. We changed "familial" with "Hereditary " as recommended by ISA.
-AA Amyloidosis, page 4, line 170. We have created Table 2, listing the main risk factors associated with autoinflammatory diseases
- "Forms of amyloidosis, page 4, line 149. I miss a paragraph here explaining why you only mention 5 types of amyloidosis. There are 36 types, why don´t you mention the other 31? Children aren’t affected?" There is inconsistent literature on other types of amyloidosis; therefore, we concentrated on the most studied forms.
-Clinical manifestations, page 6, line 226. “of an underlying disease”
-Heart, page 8, line 264. "Ventricular"
-"Diagnosis, page 9, line 310. I think you should mention what immunohistochemical typing is currently advised. You should recommend a panel of antibodies for the characterization. You should also explain why immunofluorescence is preferred over conventional immunohistochemistry" We have further explored the issue. Literature remains divided on which diagnostic technique is superior between immunohistochemistry and immunofluorescence. Nevertheless, we emphasized the matter by recommending an antibody panel.
- Treatment, page 9, line 326. Eliminating the precursors necessary for their formation.
- Treatment, page 9, line 328. A dot is required before “in fact”
-"Treatment, pages 9 and 10, lines 331-371. This paragraph should be divided, probably in 3 smaller paragraphs. One for AA amyloidosis, other for AL and a third one for TTR. You should also include a paragraph for the other less common variants, at least the ones described in section 3.2. Please remember that there are other forms of amyloidosis, which were not mentioned in the text." We have divided the section into smaller paragraphs for each form, as you indicate
- Conclusion, page 10, lines 373-376. We have eliminated these lines as you indicate
-Conclusion, page 10. We have renamed the conclusions section to "discussion"
We have also made additional modifications as suggested by Reviewer 2, including a more detailed description of the initial symptoms/signs in the two populations based on immunophenotype, as well as a revised treatment chapter that better compares adults and children.
We would like to express our gratitude for your efforts
Best Regards

Reviewer 2 Report
Comments and Suggestions for Authors
In the narrative review, the authors elucidated the main clinical characteristics of amyloidosis with onset in childhood in comparisson with the adult form. Although the findings seem to be practically useful, I would like to make some comments.
1. The authors might clearly compare the early signs / symptoms between both populations depending on immune phenotype of amyloidosis.
2. The management of the disease seems not to be comparable reported. Please, re-write this section taking into consideration the initial hypothesis of the study
Author Response
First and foremost, I would like to express my gratitude for your thorough review of our work.
Below are the corrections made as you indicated.
- "The authors might clearly compare the early signs / symptoms between both populations depending on immune phenotype of amyloidosis." We have aimed to provide a clearer description of the initial symptoms and signs in the two populations based on the immunophenotype of amyloidosis, noting that initial symptoms are often more dependent on the primarily involved organ than on the phenotype itself. We have nonetheless highlighted the differences in presentation related to the involvement of each specific organ.
- "The management of the disease seems not to be comparable reported. Please, re-write this section taking into consideration the initial hypothesis of the study" We have completely rewritten the treatment section, comparing adults and children based on the existing literature.
First and foremost, I would like to express my gratitude for your thorough review of our work.
Below are the corrections made as you indicated.
- "The authors might clearly compare the early signs / symptoms between both populations depending on immune phenotype of amyloidosis." We have aimed to provide a clearer description of the initial symptoms and signs in the two populations based on the immunophenotype of amyloidosis, noting that initial symptoms are often more dependent on the primarily involved organ than on the phenotype itself. We have nonetheless highlighted the differences in presentation related to the involvement of each specific organ.
- "The management of the disease seems not to be comparable reported. Please, re-write this section taking into consideration the initial hypothesis of the study" We have completely rewritten the treatment section, comparing adults and children based on the existing literature
We have also made additional modifications as suggested by Reviewer 1.
We would like to express our gratitude for your efforts
Best regards.
